# Environmental Regulations on the Spatial Spillover of the Sustainable Development Capability of Chinese Clustered Ports

**Xinhua He [1], Wenjun Liu [1], Ruiqi Hu [2] and Wenfa Hu [3],***

1   School of Economics and Management, Shanghai Maritime University, Shanghai 201306, China;
    xhhe@smu.edu.cn (X.H.); 17803428202@163.com (W.L.)
2   College of Letters and Science, University of Wisconsin-Madison, Madison, WI 53703, USA; rhu38@wisc.edu
3   School of Economics and Management, Tongji University, Shanghai 200092, China
*   Correspondence: wenfahu@hotmail.com; Tel.: +86-21-6598-2955

**Abstract:** For years, China has adopted environmental regulations in developing ports to improve their sustainability. Based on the data of Chinese ports from 2009 to 2018, this paper presents a data envelopment analysis model with subdividing input-output indicator weights and develops it further in two stages with the weight preference and the slacks-based measure, respectively. After assessing the sustainable development capability (SDC) of Chinese ports and their spatial correlation, it revealed that Chinese ports are clustered in several regions and their SDC has spilled over into their neighbors. Further study revealed the SDC is affected by environmental regulations in different ways: as a key measure among regulations to improve the SDC, voluntary regulation has a spatial spillover effect, but neither the mandatory regulation nor public media regulation can significantly improve the SDC. This suggests that the port authority should enact environmental regulations based on the port spatial difference and the port should expand its operation scale and market size and recruit more top talent, which is good for improving its productivity and reducing its carbon emissions.

**Keywords:** environmental regulation; sustainable development capability; ports; operation; data envelopment analysis; spatial spillover; China; cluster

## 1. Introduction

Ports are important infrastructures to support international trade. The cargo volume through Chinese ports was 14.35 billion tons in 2018, ranking first in the world. It is supposed that the Chinese cargo volume will steadily increase in the future. The development of ports is becoming a comprehensive indicator to measure the country's competitive level [1,2]. However, construction of a port demands a lot of resources, and it has caused many environmental problems in China, which have alerted the Chinese government to be concerned with environmental protection measures in the sustainable development path.

Along with the nationwide supply-side reform, China suggests developing the marine-related economy as an economical strategy. Sustainable development capability (SDC) is a key indicator to assess regional development [3]. Sustainable development is related to environmental protection including resource recycling, clean energy, and low-carbon emission, which is adjusted by environmental regulations (ERs). However, ERs have a wide influence on regional economic development, and the Porter Hypothesis is widely utilized to review the environment–competitiveness relationship [4]. Nevertheless, their relationship is so complicated that it is still not clear. Though the Porter Hypothesis has been further investigated by many researchers [5,6], a consensus has not been reached on the effect of ERs. Scholars have studied the SDC in various entities (i.e., insurance enterprises [7], electricity industries [8], and agriculture in countries [9]). When ports are

located in different cities, their SDC will show some spatial features as well as their ERs, but few papers have investigated these.

This paper aims to investigate the relationship between the SDC of Chinese ports and ERs from a spatial perspective, and is organized as follows. Section 2 reviews the literature, which builds the foundation for this research. Section 3 illustrates the data and methodology. Section 4 presents the results and discussion, and Section 5 draws conclusions and makes recommendations.

## 2. Literature Review

### 2.1. Port's Sustainable Development Capability (SDC)

Under the pressure of global environmental challenges, countries have started to develop sustainable ports. At present, if a port can achieve sustainable economic development while slowing down environmental degradation, it can be called a green port with a SDC. The difficulty in developing a sustainable port is how to achieve a clean and efficient goal for all port activities. Scholars have presented various methods to evaluate the port's sustainable performance. Park and Yeo [10] adopted factor analysis and fuzzy set to assess the greenness of Korean ports. Wan et al. [11] combined the analytic hierarchy process method and the evidence reasoning method to evaluate the development of green ports. Teerawattana and Yang [12] presented several indicators for assessing port environmental performance by the entropy method. After evaluating biomass and photovoltaic renewable energy, Balbaa and El-Amary [13] constructed a sustainable port model for the Damietta seaport.

Meanwhile, some scholars have proposed some strategies to improve the port's SDC. Tseng and Pilcher [14] conducted a quantitative analysis by the fuzzy analytic hierarchy process and suggested the critical factors in developing green ports were environmental regulation, economic regulation, workforce, and technological progress. Peng et al. [15] suggested strategies to develop green ports by developing a port carbon emission simulation model. Li et al. [16] combined the qualitative and quantitative methods to investigate the low-carbon development strategies of ports in China.

### 2.2. Effects of Environmental Regulations (ERs) on Port's SDC

ERs are supposed to accelerate the development of sustainable ports. Scholars have probed whether the ERs are proper for the development of sustainable ports. Chang and Wang [17] claimed that ERs helped to reduce the port's emissions and improve its environment. Tichavska et al. [18] suggested that the low emission depended on how ERs were implemented in ports. International agencies have also focused on marine environmental problems. For example, the International Maritime Organization assembles pollution prevention conventions to control maritime transport emissions [19], and the European Commission enacts many port development regulations (i.e., reducing carbon emissions by 40% and utilizing maritime fuel with less than 0.1% sulfur [20]). The growth of the Brazilian port industry in recent decades has profited from solid waste management regulations learned from the European experience [21]. Canadian federal agencies require that all ports must comply with the ERs (i.e., the Canadian Shipping Act, the Canadian Environmental Protection Act, and the Canadian Water Act [22]. Since there are so many kinds of ERs, how various ERs affect the port's SDC has not been fully explained.

### 2.3. Cluster of Ports

The port cluster is an economic pattern consisting of economic activities around a port, which produces a capability to boost its business competitive advantages by the clustered companies and organizations. A port cluster comprises the port authority, public organizations, and private companies related to the port operation and cargo services (i.e., cargo handling, transportation, logistics, manufacturing, and trade).

Some scholars are dedicated to exploring the factors of port clustering. Chen et al. [23] pointed out that the development potential of a port cluster depended on port cargo

throughput as well as import and export volume. Chen and Yang [24] identified industrial transfer and capacity constraints along the Maritime Silk Road as key indicators for assessing the extent of the port cluster. Dooms et al. [25] considered performance indicators of the socio-economic impact of port clusters as a key assessment system to support and enhance port clustering.

Researchers have suggested various approaches to investigate the features of a port cluster. Benito et al. [26] presented the Diamond theory to investigate the industrial clusters in the Norwegian foreign trade economy district and concluded that the industrial clusters were conducive to enhance its competitiveness and innovativeness. After investigating the Lower Mississippi port, De Langen and Visser [27] claimed that local governance and collective action would strengthen competitiveness in clustering. Zhang [28] explored the relationship between port logistics and regional economic development and revealed that the port cluster had a positive impact on its economy. Dooms [29] also confirmed that port clusters could improve sustainable competitiveness.

However, the literature on the study of the spatial clustering characteristics of a port's SDC from a quantitative perspective is limited, which needs further study.

### 2.4. Spatial Spillover of the Port's SDC

Currently, the rapid growth of spatial datasets along with the development of geographic information systems (GISs) and remote sensing technologies has made it impossible for traditional econometrics to properly explain spatial data and their effects. Spatial spillover effects, one of the most important theoretical innovations in spatial econometrics, overcome the above impediments. On the spatial spillover of a port's SDC, scholars are keen to explore the spatial spillover effects between sustainable port development and the hinterland economy. Zhao et al. [30] investigated the spatial spillover effects of the integrated development capacity of Chinese ports on the urban economy using an entropy TOPSIS (Technique for Order Preference by Similarity to an Ideal Solution) model and a spatial econometric model. Liu and Yin [31] also verified the significant spatial spillover of the port's SDC on the economic growth of hinterland cities through a spatial panel model. Wang et al. [32] found a significant spatial spillover effect of regional tertiary output as well as regional freight traffic on the port's SDC. Liang and Li [33] inspected the spatial spillover effect of cross-regional port consolidation on the port's SDC utilizing a two-zone system spatial Durbin model [34]. Bottasso et al. [35] proposed that the sustainable development of ports tended to increase local gross domestic product (GDP) and has a large positive spillover effect on the GDP of nearby areas. Regrettably, the spatial spillover effect between ERs and the port's SDC is still unproven.

### 2.5. Methods for Studying the Port's SDC

Methods for studying the port's SDC include entropy, factor analysis, fuzzy set, hierarchical analysis, and data envelopment analysis (DEA), etc. Among these methods, the DEA, based on linear programming and statistic data, is an objective method to determine the factor weights in evaluating various management performance and is a dominating model to investigate port efficiency. Tongzon [36], Itoh [37], and Cullinane et al. [38] developed DEA models to evaluate and compare the efficiency of major ports in various countries, respectively. DEA models can be easily improved by various weight methods. Chiang et al. [39] utilized a weight set obtained by a separation method to calculate the efficiency of decision-making units (DMUs) so that a multiple-objective fractional linear programming problem was transformed into single-objective linear programming. Nguyen et al. [40] presented a bootstrapped DEA to evaluate port efficiency, and Chang et al. [41] suggested a non-radial DEA model with the slacks-based measure (SBM) to estimate port efficiency. Song et al. [42] developed a hybrid weight approach by integrating the minimax optimization method and DEA to deal with the vague decision-maker problem. However, the operational risks faced by port enterprises continue to increase, so it is worthy of further

discussion to comprehensively evaluate the port's sustainable development capabilities by stages or circumstances.

### 2.6. Research Gap

In the era of embracing a low-carbon and energy-conservation economy, although there are studies related to the cluster phenomenon, sustainable development of ports, the spatial spillover of port's SDC, and even improved DEA models, they are still many limitations if applied in analyzing the Chinese clustered ports. Primarily, the previous literature has not uncovered the mystery of whether there is a spatial heterogeneous or cluster phenomenon in the SDC of Chinese ports. Second, there is still a lack of investigating the synergy effects of ERs on the port's SDC. How to realize the coordinated development between port sustainable development and ecological benefits is worth pondering. At the same time, comparing the synergy effects of various ERs on propelling the port's SDC is still unsolved. Finally, previous studies have neglected the dynamic nature of the port operation on building the DEA model.

After reviewing the previous literature, this paper builds a weight preference (WP)-SBM-DEA model with subdividing stages to evaluate the SDC of Chinese ports, and then inspects the spatial characteristics of the SDC in various Chinese ports so that the synergy effects of ERs on improving the port's SDC in a spatial perspective are investigated and the factors affecting the port's sustainability are revealed. The ERs consisting of voluntary regulations, mandatory regulations, and public media regulations are discussed respectively.

### 3. Methodology and Variables

#### 3.1. Methodology

3.1.1. The SDC Evaluation Model

A traditional DEA model can measure the comparative efficiency of a complex system with multiple inputs and outputs, but it does not take into account the slackness of input and output and the undesired output of port production and operation activities, so it is not suitable to discern the spatial differences in their DMUs. Tone [43] developed the SBM-DEA by introducing relaxation variables in the objective functions with the slacks-based measure in the DEA, but the author ignored the difference in indicator weights and internal structure of input-output. After reviewing previous studies on sustainable development of ports [44–46], we developed a DEA model to evaluate ports that consisted of two stages, as shown in Figure 1.

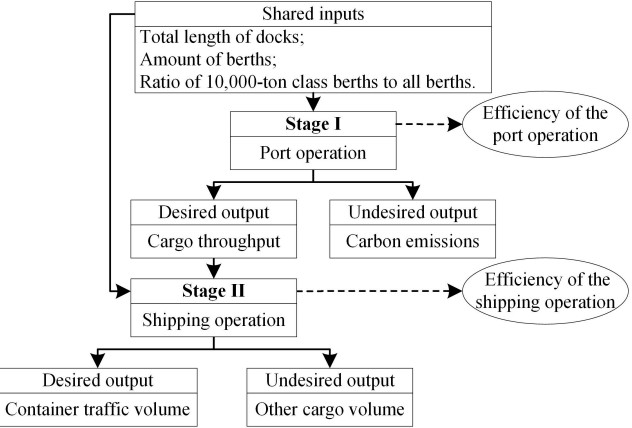

**Figure 1.** A two-stage data envelopment analysis (DEA) model for evaluating the port's sustainable development capability (SDC).

The input indicators in the model are the port size, the quantity of dock equipment, the berth length, and number of employees. Besides the input indicators, the port's capital,

workforce, and lands were chosen as the critical indicators in evaluating a port. The output indicator in the model is the annual cargo volume or annual container throughput.

The first stage is the port operation stage. To evaluate the sustainable efficiency of the port operation, the desired output is the cargo throughput and the undesired output is the carbon emissions. The second stage is the cargo operation stage. The desired output in the first phase is the intermediate input of the second phase, the container traffic volume is regarded as the desired output, and other cargo volume is regarded as the undesired output in evaluating the efficiency of the port's containerization. Besides the terminal length, the number of berths and the ratio of the 10,000-ton berths to all berths are regarded as shared inputs in the first and second phases.

To account for the slackness of various inputs and outputs in port operation, a two-stage WP-SBM-DEA model is presented by utilizing weight preference in the SBM-DEA.

It is assumed that the distribution factor $\tau$ denotes the proportion of shared inputs allocated to the first stage, and $1 - \tau$ denotes the proportion of shared inputs allocated to the second stage. $\tau$ is usually 0.5.

The two-stage WP-SBM-DEA model is defined as follows, and the subscripts 1 and 2 denote the first stage and the second stage, respectively.

In the first stage:

$$min \; \rho_{1j}^{m*} = \frac{1 - \frac{1}{a_1} \sum\limits_{r_1=1}^{a_1} \frac{w_{r_1}^{1g}}{\tau x_{r_1 j}^{1m}}}{1 + \frac{1}{b_1+c_1} \left( \sum\limits_{r_2=1}^{b_1} \frac{\omega_{r_2}^{1h}}{y_{r_2 j}^{1m}} + \sum\limits_{r_3=1}^{c_2} \frac{\omega_{r_3}^{1s}}{z_{r_3 j}^{1m}} \right)} \tag{1}$$

$$s.t. \begin{cases} \tau x_{r_1 j}^{1m} = \sum\limits_{j=1}^{k} \tau x_{r_1 j}^{1m} \lambda_j - \omega_{r_1}^{1g} \\ y_{r_2 j}^{1m} = \sum\limits_{j=1}^{k} y_{r_2 j}^{1m} \mu_j - \omega_{r_2}^{1h} \\ z_{r_3 j}^{1m} = \sum\limits_{j=1}^{k} z_{r_3 j}^{1m} \xi_j + \omega_{r_3}^{1s} \\ w_{r_1}^{1g}, \omega_{r_2}^{1h}, w_{r_3}^{1s} \geq 0; \lambda_j, \; \mu_j, \xi_j \geq 0; x_{r_1 j}^{1m}, \; y_{r_2 j}^{1m}, \; z_{r_3 j}^{1m} \geq 0 \\ r_1 = 1, 2, \ldots, a_2; r_2 = 1, 2, \ldots, b_2; r_3 = 1, 2, \ldots, c_2; j = 1, 2, \ldots, k \end{cases}$$

In the second stage:

$$min \; \rho_{2j}^{m*} = \frac{1 - \frac{1}{a_2} \sum\limits_{r_1=1}^{a_2} \frac{\omega_{r_1}^{2g}}{x_{r_1 j}^{2m}}}{1 + \frac{1}{b_2+c_2} \left( \sum\limits_{r_2=1}^{b_2} \frac{\omega_{r_2}^{2h}}{y_{r_2 j}^{2m}} + \sum\limits_{r_3=1}^{c_2} \frac{\omega_{r_3}^{2s}}{z_{r_3 j}^{2m}} \right)} \tag{2}$$

$$s.t. \begin{cases} x_{r_1 j}^{2m} = (1 - \tau) x_{r_1 j}^{1m} + y_{r_2 j}^{1m} = \sum\limits_{j=1}^{k} \left[ (1-\tau) x_{r_1 j}^{1m} + y_{r_2 j}^{1m} \right] \lambda_j + \omega_{r_1}^{2g} \\ y_{r_2 j}^{2m} = \sum\limits_{j=1}^{k} y_{r_2 j}^{2m} \mu_j - \omega_{r_2}^{2h} \\ z_{r_3 j}^{2m} = \sum\limits_{j=1}^{k} z_{r_3 j}^{2m} \xi_j + \omega_{r_3}^{2s} \\ \frac{1}{a_2} \sum\limits_{r_1=1}^{a_2} \frac{\omega_{r_1}^{2g}}{x_{r_1 j}^{2m}} = \frac{1}{a_1+b_1} \left( \sum\limits_{r_1=1}^{a_1} \frac{\omega_{r_1}^{1g}}{(1-\tau) x_{r_1 j}^{1m}} + \sum\limits_{r_2=1}^{b_1} \frac{\omega_{r_2}^{1h}}{y_{r_2 j}^{1m}} \right) \\ w_{r_1}^{2g}, \omega_{r_2}^{2h}, w_{r_3}^{2s} \geq 0; \lambda_j, \; \mu_j, \xi_j \geq 0; x_{r_1 j}^{2m}, \; y_{r_2 j}^{2m}, \; z_{r_3 j}^{2m} \geq 0 \\ r_1 = 1, 2, \ldots, a_2; r_2 = 1, 2, \ldots, b_2; r_3 = 1, 2, \ldots, c_2; j = 1, 2, \ldots, k \end{cases}$$

where $\rho_{1j}^{m*}$ represents the sustainable operation efficiency of a port in the $m^{th}$ year; $\rho_{2j}^{m*}$ represents the containerization efficiency of a port in the $m^{th}$ year; $\omega_{r_1}^{g}$, $\omega_{r_2}^{h}$ and $\omega_{r_3}^{s}$ are vectors denoting the slacks of the input indicator, the desired output indicator, and the undesired output indicator at a port $j$, respectively, where $\omega_{r_1}^{g}$ is the input excess, $\omega_{r_2}^{h}$ is the shortfall of the desired output, and $\omega_{r_3}^{s}$ is the superscalar of the undesired output; $\omega_{r_1}^{1g}/\tau x_{r_1j}^{1m}$ is the input redundancy ratio in the first stage; $\omega_{r_1}^{2g}/x_{r_1j}^{2m}$ is the input redundancy ratio in the second stage; $\omega_{r_2}^{h}/y_{r_2j}^{m}$ is the desired output redundancy; and $\omega_{r_3}^{s}/z_{r_3j}^{m}$ is the undesired output redundancy.

$\rho_j^{m*}$ is a strict monotonic decreasing function, $0 \leq \rho_j^{m*} \leq 1$. Supposing the optimal solution is represented by $\left(\lambda_{r_1}^{*}, \mu_{r_2}^{*}, \xi_{r_3}^{*}, w_{r_1}^{g}{}^{*}, \omega_{r_2}^{h*}, w_{r_3}^{s*}\right)$, when $\rho_j^{m*} = 1$, $\omega_{r_1}^{g}{}^{*} = 0$, $\omega_{r_2}^{h*} = 0$, and $\omega_{r_3}^{s*} = 0$, the solution is optimally efficient. When $\rho_j^{m*} < 1$, the port can improve its SDC by adjusting the values of input indicators, desired output indicators, and undesired output indicators, so that $\rho_j^{m*} \to 1$.

$$\phi_1 = \frac{\sum_{j=1}^{k} \tau x_{r_1j}^{m}\lambda_j + \sum_{j=1}^{k} y_{r_2j}^{1m}\mu_j + \sum_{j=1}^{k} z_{r_3j}^{1m}\xi_j}{\sum_{j=1}^{k} x_{r_1j}^{1m}\lambda_j + \sum_{j=1}^{k} y_{r_2j}^{1m}(\lambda_j + \mu_j) + \sum_{j=1}^{k} y_{r_2j}^{2m}\mu_j + \sum_{j=1}^{k} \left(z_{r_3j}^{1m} + z_{r_3j}^{2m}\right)\xi_j} \tag{3}$$

$$\phi_2 = \frac{\sum_{j=1}^{k} \left[(1-\tau)x_{r_1j}^{1m} + y_{r_2j}^{1m}\right]\lambda_j + \sum_{j=1}^{k} y_{r_2j}^{2m}\mu_j + \sum_{j=1}^{k} z_{r_3j}^{2m}\xi_j}{\sum_{j=1}^{k} x_{r_1j}^{1m}\lambda_j + \sum_{j=1}^{k} y_{r_2j}^{1m}(\lambda_j + \mu_j) + \sum_{j=1}^{k} y_{r_2j}^{2m}\mu_j + \sum_{j=1}^{k} \left(z_{r_3j}^{1m} + z_{r_3j}^{2m}\right)\xi_j} \tag{4}$$

$$\Omega_j^m = \phi_1 \rho_{1j}^{m*} + \phi_2 \rho_{2j}^{m*} \tag{5}$$

where $\phi_1$ and $\phi_2$ represent the weights in the two stages, respectively. $\sum_{j=1}^{k} x_{r_1j}^{1m}\lambda_j + \sum_{j=1}^{k} y_{r_2j}^{1m}(\lambda_j + \mu_j) + \sum_{j=1}^{k} y_{r_2j}^{2m}\mu_j + \sum_{j=1}^{k} \left(z_{r_3j}^{1m} + z_{r_3j}^{2m}\right)\xi_j$ denotes the total amount of input and output in the two-stage network DEA model. $\sum_{j=1}^{k} \tau x_{r_1j}^{m}\lambda_j + \sum_{j=1}^{k} y_{r_2j}^{1m}\mu_j + \sum_{j=1}^{k} z_{r_3j}^{1m}\xi_j$ and $\sum_{j=1}^{k} \left[(1-\tau)x_{r_1j}^{1m} + y_{r_2j}^{1m}\right]\lambda_j + \sum_{j=1}^{k} y_{r_2j}^{2m}\mu_j + \sum_{j=1}^{k} z_{r_3j}^{2m}\xi_j$ denote the total amount of input and output in the first and second stages, respectively. $\Omega_j^m$ represents the SDC of port $j$ in the $m^{th}$ year.

### 3.1.2. Spatial Correlation Test of the Port's SDC

A spatial autocorrelation model is often used to test the degree of correlation between adjacent regions and to discern the spatial correlation, which is the most popular global cluster analysis method, also known as the Moran's I test. This method is adjusted to investigate the spatial relationship of Chinese ports, defined by Equation (6) [47]:

$$I = \frac{n \sum_{i=1}^{n} \sum_{j=1}^{n} W_{ij}\left(X_i - \overline{X}\right)\left(X_j - \overline{X}\right)}{\left(n \sum_{i=1}^{n} \sum_{j=1}^{n} W_{ij}\right) \sum_{i}^{n}\left(X_i - \overline{X}\right)^2} \tag{6}$$

where $I$ denotes the value of Moran's $I$, ranging from $-1$ to 1. $I > 0$ signifies an affirmative spatial autocorrelation on the SDC, and $I < 0$ signifies an unfavorable spatial autocorrelation; $\overline{X} = \frac{1}{n} \sum_{i=1}^{n} X_i$, and $X_i$ is the SDC of port $i$; $W_{ij}$ is the spatial weight matrix, describing the correlation effect of the spatial dependence and heterogeneity of an observed variable, and verifying the spatial spillover effect.

The z-statistic $Z(I)$, after standardizing the Moran's I, is chosen to inspect its significance, fitting a standard normal distribution asymptotically [48,49].

$$Z(I) = (I - E(I)) / \sqrt{V(I)} \tag{7}$$

where $E(I) = -1/(n-1)$ and $V(I) = E(I^2) - E(I)$.

After reviewing the previous literature [50,51], the spatial weight matrix $W_{ij}$ is further represented by the adjacent matrix $W_{ij}^A$, the geospatial distance matrix $W_{ij}^s$, and the economic distance matrix $W_{ij}^E$, respectively. $W_{ij}^A$ reflects the spatial adjacent relationship between ports, $W_{ij}^s$ is the geospatial proximity between ports, and $W_{ij}^E$ represents the gap in economic development between ports.

$$W_{ij}^A = \begin{cases} 1 \ (\text{two ports i and j are adjacent}) \\ 0 \ (two\ ports\ i\ and\ j\ are\ not\ adjacent) \end{cases} \tag{8}$$

$$W_{ij}^s = \begin{cases} 1/d_{ij} \ (i \neq j) \\ 0 \ (i = j) \end{cases} \tag{9}$$

$$W_{ij}^E = \begin{cases} 1/|\overline{Y}_i - \overline{Y}_j| \ (i \neq j) \\ 0 \ (i = j) \end{cases} \tag{10}$$

where $d_{ij} = r \cos^{-1}\left[\cos(E_i - E_j) \cos N_i \cos N_j + \sin N_i \sin N_j\right]$; r is the earth radius ; $E_i$ is the longitude of port $i$; $N_i$ is the latitude of port $i$; and $\overline{Y}_i$ and $\overline{Y}_j$ are the average annual revenue of port $i$ and j, respectively.

### 3.1.3. Space Panel Econometric Models

There are three types of spatial panel econometric models [52,53]: the spatial panel lag model (SAR), the spatial panel error model (SEM), and the spatial panel Durbin model (SDM). The panel model for assessing the port's SDC is developed by the following steps:

(1) A ordinary least squares (OLS) regression is used to evaluate the port's panel data.

$$C_{im} = \alpha_i + \beta X_{im} + e_{im} \tag{11}$$

where $C_{im}$ is the SDC of port $i$ at time $m$; $\alpha_i$ is a constant; $\beta$ is a coefficient ; $X_{im}$ is the set of independent variables including the explanatory variables and control variables; and $e_{im}$ is a tiny error term.

(2) A spatial effect $\omega_s$ and a time effect $\varepsilon_t$ are introduced to the OLS regression model, representing spatial changes over time, which makes a spatial econometric model.

$$C_{im} = \alpha_i + \beta X_{im} + e_{im} + \omega_s + \varepsilon_t \tag{12}$$

(3) A spatial weight matrix $W_{ij}$ is introduced to the spatial econometric model so that an integrated spatial model is presented, where $W_{ij}$ is $W_{ij}^A$, $W_{ij}^s$ or $W_{ij}^E$, denoting the adjacency matrix, the geospatial distance matrix, and the economic distance matrix.

$$C_{im} = \alpha_i + \beta X_{im} + \eta \sum_{j=1}^{n} W_{ij} C_{jm} + \delta \sum_{j=1}^{n} W_{ij} \varphi_{jm} + \gamma \sum_{j=1}^{n} W_{ij} X_{jm} + \sigma C_{i,m-1} + e_{im} + \omega_s + \varepsilon_t$$
$$\tag{13}$$

where $\eta$ is a coefficient representing the spatial spillover effect of SDC; $\delta$ is a coefficient representing the spatial spillover effect of the error term; $\varphi$ is the spatial autocorrelation error term; $\gamma$ is a coefficient representing the spatial spillover effect of ERs; $C_{i,m-1}$ is the explained variable with a first-order lag; $\sigma$ is a coefficient; and $i$ and $j$ are two ports ($i = 1, 2, \ldots, n$; $j = 1, 2, \ldots, n$).

Three types of the model are defined as follows: When $\delta = 0$, Equation (13) is an SDM model; when $\delta = 0$, $\gamma = 0$, it is an SLM model; when $\sigma = 0$, $\eta = 0$, and $\delta = 0$, it is an SEM model.

### 3.2. Data and Variables

#### 3.2.1. Data Sources

The input and output data are from various sources including the China Statistical Yearbook, China Environmental Yearbook, and China's Port Statistical Yearbook. The Chinese ports are classified by their regions (shown in Table 1), and the port's SDC is set as the explained variable.

**Table 1.** Chinese ports and their regions.

| Region | Port Name |
| --- | --- |
| Bohai Rim Region | Dalian Port, Yingkou Port, Tianjin Port, Tangshan Port, Qinhuangdao Port, Qingdao Port, Yantai Port, Rizhao Port |
| Yangtze River Delta Region | Shanghai Port, Ningbo Port, Lianyungang Port |
| Southeast Coastal Area | Xiamen Port, Fuzhou Port |
| Pearl River Delta Region | Guangzhou Port, Shenzhen Port, Zhuhai Port, Shantou Port |
| Southwest Coastal Area | Zhanjiang Port, Beibu Gulf Port, Haikou Port |

#### 3.2.2. Explanatory Variables

There are several methods to select explanatory variables in the Porter Hypothesis. Majumdar and Marcus [54] and Tosun and Knill [55] divided ERs into agile regulations and stiff regulations. Chris et al. [56] divided ERs into voluntary regulations and mandatory regulations. After reviewing the literature, we divided ERs into mandatory regulations, public media regulations, and voluntary regulations.

A mandatory regulation enables the port authority to monitor whether the port operation meets the environmental standards or not. Investment in environmental pollution control (EPC) was utilized as an indicator of mandatory regulation.

Public media could be a social monitor who is aware of the port environmental behaviors and pollution problems. After an environmental pollution event happens, the port managers have to deal with the pressure from the public media, and the port would lose its market share [57]. Port market share (PMS) was set as an indicator for public media regulations.

Voluntary regulations are the additional measures that a port adopts voluntarily for environmental protection during its operation. Environmental protection expenditure (EPE) was set as an indicator of voluntary regulations.

#### 3.2.3. Control Variables

After reviewing previous studies [58,59], we suggested five control variables: (1) operation scale, (2) market size, (3) international trade dependence, (4) knowledge accumulation capacity, and (5) environmental pollution.

The operation scale can improve the competitiveness of a port, and the gross operation income (GOI) was chosen to represent its operation scale. The market size of a port is affected by the local population (POP). A port with a higher SDC may have a higher international trade volume, and the ratio of the foreign trade volume to the domestic one (FTD) is to denote the international trade dependence. Knowledge helps to develop innovative ideas and improve port productivity, and the knowledge accumulation capacity (KAC) is denoted by the proportion of employees holding a bachelor's degree or above to the total employees in a port. Environmental pollution is usually determined by carbon emissions ($CO_2$) in a port, estimated by a method proposed by Liao et al. [60], which is determined by the cargo volume and energy consumption of ports.

## 4. Results

### 4.1. SDC of Twenty Chinese Ports

Descriptive statistical data of variables from 20 Chinese ports are listed in Table 2.

**Table 2.** Descriptive statistical data of the variables.

| Indicator | Variable | Unit | Mean | Max | Min | S.D. |
|---|---|---|---|---|---|---|
| | Dock length | m | 31,109.2 | 126,921 | 4563 | 26,825.71 |
| The shared Inputs | Quantity of berths | count | 229.755 | 1238 | 32 | 273.741 |
| | Ratio of 1000-ton berths to all berths | % | 43.97 | 94.3 | 2 | 25.24 |
| Desired outputs | Cargo volume | $\times 10^8$ ton | 3.043 | 10.84 | 0.23 | 2.077 |
| | Container traffic volume | $\times 10^4$ TEU | 800.54 | 4201 | 20.56 | 933.91 |
| Undesired outputs | Other cargo volume | $\times 10^8$ ton | 1.54 | 7.95 | 0.02 | 1.73 |
| | $CO_2$ | $\times 10^4$ ton | 22.59 | 78.64 | 1.48 | 13.87 |
| Explained variables | SDC | / | 0.617 | 0.953 | 0.065 | 0.198 |
| | EPC | $\times 10^8$ yuan | 52.88 | 278.9 | 6.9 | 49.4 |
| Explanatory variables | EPE | $\times 10^8$ yuan | 22.73 | 233.39 | 0.79 | 27.51 |
| | PMS | % | 33.34 | 66.3 | 8.6 | 11.83 |
| | GOI | $\times 10^8$ yuan | 61.24 | 380.43 | 2.16 | 67.06 |
| | POP | $\times 10^4$ individual | 735.2 | 2426 | 149.1 | 513.2 |
| Control variables | FTD | % | 40.57 | 48.8 | 32.7 | 5.88 |
| | KAC | % | 20.71 | 49.6 | 3 | 10.3 |
| | $CO_2$ | $\times 10^4$ ton | 22.59 | 78.64 | 1.48 | 13.87 |

The SDC of ports was evaluated by the two-stage WP-SBM-DEA model, and their SDC values from 2009 to 2018 with their means shown in Table 3.

**Table 3.** Sustainable development capability (SDC) of 20 Chinese ports from 2009 to 2018.

| Port No. | Port Name | 2009 | 2010 | 2011 | 2012 | 2013 | 2014 | 2015 | 2016 | 2017 | 2018 | Mean |
|---|---|---|---|---|---|---|---|---|---|---|---|---|
| 1 | Dalian Port | 0.710 | 0.730 | 0.720 | 0.743 | 0.754 | 0.733 | 0.750 | 0.754 | 0.761 | 0.767 | 0.742 |
| 2 | Tianjin Port | 0.854 | 0.844 | 0.856 | 0.885 | 0.879 | 0.884 | 0.881 | 0.888 | 0.876 | 0.895 | 0.874 |
| 3 | Shanghai Port | 0.923 | 0.933 | 0.924 | 0.942 | 0.945 | 0.926 | 0.947 | 0.953 | 0.941 | 0.963 | 0.940 |
| 4 | Ningbo Port | 0.895 | 0.901 | 0.898 | 0.894 | 0.892 | 0.864 | 0.852 | 0.902 | 0.852 | 0.865 | 0.882 |
| 5 | Guangzhou Port | 0.837 | 0.845 | 0.846 | 0.842 | 0.836 | 0.812 | 0.806 | 0.852 | 0.806 | 0.813 | 0.830 |
| 6 | Shenzhen Port | 0.755 | 0.763 | 0.616 | 0.639 | 0.672 | 0.688 | 0.702 | 0.719 | 0.717 | 0.727 | 0.700 |
| 7 | Qingdao Port | 0.803 | 0.802 | 0.827 | 0.794 | 0.800 | 0.806 | 0.799 | 0.811 | 0.783 | 0.793 | 0.802 |
| 8 | Xiamen Port | 0.681 | 0.680 | 0.680 | 0.661 | 0.633 | 0.688 | 0.663 | 0.678 | 0.672 | 0.673 | 0.671 |
| 9 | Tangshan Port | 0.634 | 0.637 | 0.640 | 0.627 | 0.615 | 0.655 | 0.627 | 0.632 | 0.633 | 0.630 | 0.633 |
| 10 | Qinhuangdao Port | 0.589 | 0.592 | 0.593 | 0.586 | 0.584 | 0.573 | 0.591 | 0.584 | 0.594 | 0.587 | 0.587 |
| 11 | Yingkou Port | 0.650 | 0.648 | 0.655 | 0.640 | 0.637 | 0.633 | 0.652 | 0.656 | 0.627 | 0.644 | 0.644 |
| 12 | Lianyungang Port | 0.570 | 0.568 | 0.613 | 0.616 | 0.614 | 0.609 | 0.617 | 0.614 | 0.593 | 0.592 | 0.601 |
| 13 | Rizhao Port | 0.516 | 0.529 | 0.535 | 0.524 | 0.529 | 0.532 | 0.536 | 0.531 | 0.527 | 0.509 | 0.527 |
| 14 | Zhanjiang Port | 0.476 | 0.498 | 0.505 | 0.509 | 0.503 | 0.492 | 0.507 | 0.502 | 0.497 | 0.479 | 0.497 |
| 15 | Beibu Gulf Port | 0.416 | 0.433 | 0.434 | 0.446 | 0.431 | 0.429 | 0.426 | 0.418 | 0.421 | 0.418 | 0.427 |
| 16 | Fuzhou Port | 0.532 | 0.556 | 0.565 | 0.553 | 0.550 | 0.543 | 0.521 | 0.486 | 0.490 | 0.496 | 0.529 |
| 17 | Yantai Port | 0.546 | 0.568 | 0.579 | 0.564 | 0.561 | 0.552 | 0.53 | 0.5 | 0.522 | 0.512 | 0.543 |
| 18 | Zhuhai Port | 0.385 | 0.428 | 0.420 | 0.436 | 0.442 | 0.431 | 0.433 | 0.423 | 0.438 | 0.435 | 0.427 |
| 19 | Shantou Port | 0.352 | 0.394 | 0.381 | 0.379 | 0.389 | 0.366 | 0.374 | 0.369 | 0.374 | 0.368 | 0.375 |
| 20 | Haikou Port | 0.065 | 0.068 | 0.066 | 0.087 | 0.113 | 0.127 | 0.139 | 0.144 | 0.158 | 0.144 | 0.111 |

Notes: The data were obtained by MATLAB R2018b. Original data were retrieved from the China Statistical Yearbook, China Environmental Yearbook, and China's Port Statistical Yearbook.

After reviewing their locations, the result revealed that ports in the Bohai Rim Region and the Yangtze River Delta Region had higher SDC than other regions. Though most ports had a steady tendency of improving their SDC over time, the gaps of their SDC did not narrow. Chinese ports were divided into six groups based on their SDC values (shown in Table 4).

**Table 4.** Classification of Chinese ports by their SDC.

| Grade | Range | Port No. | Count of Ports |
|-------|-------|----------|----------------|
| 1 | $0.9 \leq SDC \leq 1$ | 3 | 1 |
| 2 | $0.8 \leq SDC < 0.9$ | 2, 4, 5, 7 | 4 |
| 3 | $0.7 \leq SDC < 0.8$ | 1, 6 | 2 |
| 4 | $0.6 \leq SDC < 0.7$ | 8, 9, 11, 12 | 4 |
| 5 | $0.5 \leq SDC < 0.6$ | 10, 13, 16, 17 | 4 |
| 6 | $0 \leq SDC < 0.5$ | 14, 15, 18, 19, 20 | 5 |

The Shanghai port is the only one ranked Grade 1, whose SDC was 0.939. One-quarter were bigger than 0.8, which are located in megacities. Nearly half of Chinese ports were lower than 0.6, which are mainly located in middle-size cities along the Chinese east coastline. This result shows that the majority of Chinese ports need to improve their SDC.

*4.2. Spatial Autocorrelation Test*

4.2.1. Global Spatial Autocorrelation Analysis

The global spatial autocorrelation of Chinese ports in their SDC is denoted by the Moran's I values, evaluated by $W_{ij}^A$, $W_{ij}^S$, and $W_{ij}^E$ and shown in Table 5.

The values of Moran's I of Chinese ports has remained stable over the last decade. Though the Moran's I value determined by the spatial adjacency matrix decreased slightly in the last three years, the other two Moran's I values remained unchanged from 2009 to 2018. Considering the Chinese GDP was 5.1 trillion USD in 2009 and 13.6 trillion USD in 2018, the Moran's I value by the spatial economic distance matrix remained unchanged, which was 0.347 in 2018 and 0.355 in 2009.

**Table 5.** Global spatial autocorrelation of Chinese ports from 2009 to 2018.

| Year | 2009 | 2010 | 2011 | 2012 | 2013 |
|------|------|------|------|------|------|
| *Moran's I by* $W_{ij}^A$ | 0.235 ** (1.659) | 0.238 ** (1.699) | 0.262 ** (1.848) | 0.248 ** (1.753) | 0.262 ** (1.814) |
| *Moran's I by* $W_{ij}^S$ | 0.053 * (1.387) | 0.054 * (1.422) | 0.086 ** (1.851) | 0.075 ** (1.691) | 0.077 ** (1.704) |
| *Moran's I by* $W_{ij}^E$ | 0.355 *** (3.174) | 0.340 *** (3.096) | 0.313 ** (2.897) | 0.312 ** (2.870) | 0.306 ** (2.797) |
| **Year** | **2014** | **2015** | **2016** | **2017** | **2018** |
| *Moran's I by* $W_{ij}^A$ | 0.253 ** (1.754) | 0.238 ** (1.662) | 0.191 * (1.376) | 0.214 * (1.510) | 0.212 * (1.499) |
| *Moran's I by* $W_{ij}^A$ | 0.074 ** (1.659) | 0.068 * (1.576) | 0.049 * (1.310) | 0.06 * (1.455) | 0.06 * (1.448) |
| *Moran's I by* $W_{ij}^A$ | 0.343 *** (3.067) | 0.329 ** (2.948) | 0.331 ** (2.924) | 0.351 *** (3.092) | 0.347 *** (3.054) |

Note: Z-statistics in the parenthesis. *, **, *** mean that *p* values are less than 0.1, 0.05, and 0.01, respectively. All tables below are the same. The data were obtained by Stata 15.0.

4.2.2. Local Spatial Autocorrelation Analysis

The Moran's I values of Chinese ports in their SDC by $W_{ij}^A$ and $W_{ij}^S$ passed the 10% significance and by $W_{ij}^E$ passed the 5% significance test, which demonstrates that the Chinese ports' SDC was significantly dependent on the economic distance. Chinese ports were divided into four groups: H-H, L-H, L-L, and H-L, according to their SDC by $W_{ij}^E$ in 2009, 2012, 2015, and 2018 (shown in Table 6), where H represents a higher correlation and L represents a lower correlation.

**Table 6.** Distribution of ports based on their SDC.

| Year | Correlation Mode | Port No. | Port Quantity |
|------|------------------|----------|---------------|
| 2009 | H-H | 1, 2, 3, 4, 5, 6, 7, 8, 9 | 9 |
|      | L-H | 10, 13 | 2 |
|      | L-L | 12, 14, 15, 16, 17, 18, 19, 20 | 8 |
|      | H-L | 11 | 1 |
| 2012 | H-H | 1, 2, 3, 4, 5, 6, 7, 8, 9 | 9 |
|      | L-H | 10, 13,18 | 3 |
|      | L-L | 12, 14, 15, 16, 17, 19, 20 | 7 |
|      | H-L | 11 | 1 |
| 2015 | H-H | 1, 2, 3, 4, 5, 6, 7, 8, 9 | 9 |
|      | L-H | 10,18 | 2 |
|      | L-L | 12, 13,14, 15, 16, 17, 19, 20 | 8 |
|      | H-L | 11 | 1 |
| 2018 | H-H | 1, 2, 3, 4, 5, 6, 7, 8, 9 | 9 |
|      | L-H | 10 | 1 |
|      | L-L | 12, 13, 14, 15, 16, 17, 18, 19, 20 | 9 |
|      | H-L | 11 | 1 |

Out of 20 ports over the last decade, nine ports were in H-H, whose SDC were higher and had a higher cluster, and seven to nine ports were in the L-L, whose SDC were lower and their gaps in economic development were small. There are only one to three ports located in L-H, which had a lower cluster degree, but were surrounded by ports with higher SDC. One port, Yingkou Port, remained in H-L from 2009 to 2018, where the SDC was higher, but is surrounded by ports with lower SDC.

The local indicators of spatial association (LISA) are often used to reflect the spatial aspects [61]. Their LISAs are visualized in five regions with their geographic locations, as shown in Figure 2. Their cluster maps are visualized in Figure 3, which also validates the spatial heterogeneity and cluster phenomenon of the port's SDC.

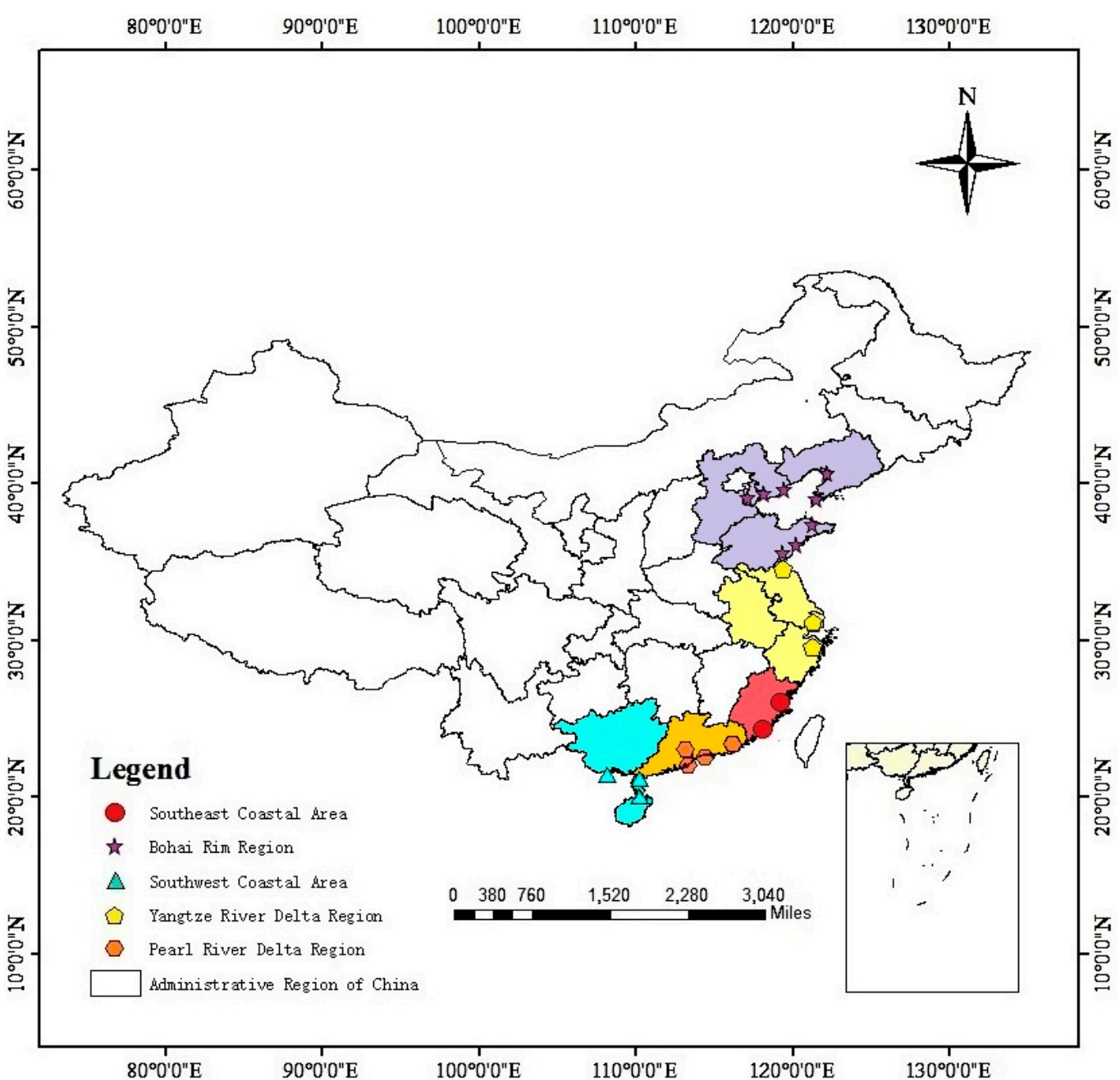

**Figure 2.** Spatial locations of the Chinese 20 ports in five regions.

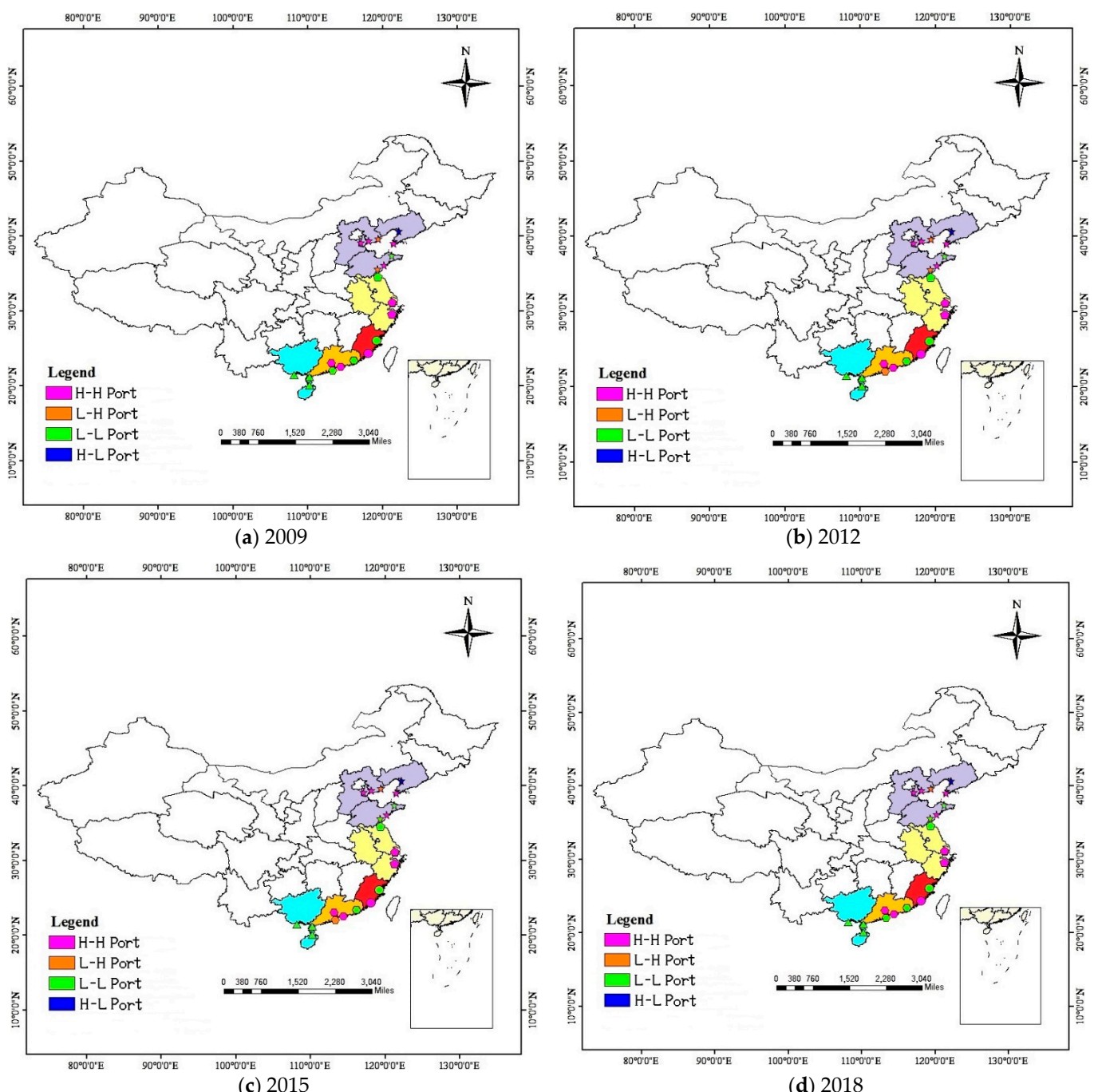

**Figure 3.** Spatial relationships of twenty Chinese ports, 2009–2018. The figures were drawn by ArcGIS 10.3.

*4.3. Results by Various Econometric Models*

The SDC of Chinese ports were investigated by conventional econometric models with OLS, fixed-effect, and random-effect, respectively, and spatial panel econometric models with $W_{ij}^A$, $W_{ij}^S$, and $W_{ij}^E$, respectively.

4.3.1. Conventional Panel Econometric Models

Several conventional panel econometric models with OLS, fixed-effect, and random-effect were constructed as the baseline. Their Hausman test was 22.81 ($p < 0.01$), revealing that the econometric models are suitable to analyze the fixed-effects of ports. Their variance-inflating factors (VIFs) were small, which showed no multi-collinearity between indicators (Table 7).

**Table 7.** Results of the conventional panel econometric models.

| Variable | OLS Model | Fixed-Effect Model | Random-Effect Model | VIFs |
|---|---|---|---|---|
| EPC | −0.48 (−0.51) | −0.055 (−0.74) | −0.056 (−0.72) | 1.97 |
| EPE | 0.707 (1.26) | 0.472 *** (10.69) | 0.472 *** (10.21) | 1.21 |
| PMS | −0.702 ** (−2.32) | 0.022 (0.88) | 0.021 (0.8) | 1.07 |
| GOI | 0.462 (0.84) | 0.035 (0.81) | 0.036 (0.78) | 2.33 |
| POP | −1.102 (−0.22) | 0.282 (0.71) | 0.28 (0.68) | 2.71 |
| FTD | 0.043 (0.05) | 0.076 (1.16) | 0.076 (1.1) | 2.76 |
| KAC | 0.043 (0.06) | 0.124 ** (2.15) | 0.124 ** (2.06) | 1.62 |
| $CO_2$ | 0.403 (0.41) | 0.122 (1.57) | 0.122 (1.5) | 1.19 |
| sigma$\_u$ | | 0.201 | 0.143 | |
| $R^2$ | 0.035 | 0.441 | 0.441 | |
| Obs | 200 | 200 | 200 | |
| Hausman-test | | 22.81 *** | | |

Note: t-statistics in parenthesis. **, *** mean that *p* values are less than 0.05, and 0.01, respectively. All tables below are the same. The data were obtained by Stata 15.0.

### 4.3.2. Spatial Panel Econometric Model

Among the three types of spatial panel econometric model, the Hausman test suggests that SDM should choose the fixed-effects, where the fitting goodness $R^2$ in the space fixed-effect model is the largest, and the log-likelihood value in the spatial-temporal fixed-effect model is the largest. Therefore, the SDM is suitable for analyzing Chinese ports, and the spatial fixed-effect model or the spatial-temporal fixed-effect model is recommended for the SDM. Three regressional results of Chinese ports by three spatial econometric models with $W_{ij}^A$, $W_{ij}^S$, and $W_{ij}^E$ are shown in Tables 8–10, respectively.

The regressional results by spatial econometric models with $W_{ij}^A$ showed that four variables, EPE, POP, FTD, and KAC, were positive and statistically significant both in the spatial fixed-effect model and the spatial-temporal fixed-effect model. The results also revealed that three spatial lag variables, $W^*$EPE, $W^*$FTD, and $W^*$KAC, were statistically significant in the spatial fixed-effect model, and two spatial lag variables, $W^*$PMS, and $W^*$KAC, were statistically significant in the spatial-temporal fixed-effect model.

The regressional results by spatial econometric models with $W_{ij}^S$ showed that three variables, EPE, POP, and FTD, were positive and statistically significant in the spatial fixed-effect model, and six variables, EPE, GOI, POP, FTD, KAC, and $CO_2$, were positive and statistically significant in the spatial-temporal fixed-effect model. The results also showed that three spatial lag variables, $W^*$GOI, $W^*$FTD, and $W^*$KAC, were statistically significant in the spatial fixed-effect model, and three spatial lag variables, $W^*$PMS, $W^*$FTD, and $W^*$CO$_2$, were statistically significant in the spatial-temporal fixed-effect model.

The regressional results by spatial econometric models with $W_{ij}^E$ showed that two variables, EPE and KAC, were positive and statistically significant in the spatial fixed-effect model, and four variables, EPE, GOI, POP, and FTD, were positive and statistically significant in the spatial-temporal fixed-effect model. The results also showed that two spatial lag variables, $W^*$EPE and $W^*$POP, were statistically significant in the spatial fixed-

effect model, and one spatial lag variable, $W^*$EPE, was statistically significant in the spatial-temporal fixed-effect model.

**Table 8.** Regressional results by spatial econometric models with $W_{ij}^A$.

| Variable | SAR | SEM | SDM | | | |
|---|---|---|---|---|---|---|
| | | | No Fixed | Spatial Fixed | Time Fixed | Spatial-Temporal Fixed |
| EPC | −0.059 (−0.84) | −0.071 (−1.01) | 0.035 (0.44) | 0.035 (0.48) | −0.121 (−0.11) | 0.003 (0.04) |
| EPE | 0.473 *** (11.51) | 0.484 *** (11.68) | 0.515 *** (12.13) | 0.515 *** (12.77) | 0.812 * (1.39) | 0.519 *** (12.57) |
| PMS | 0.022 (0.94) | 0.017 (0.72) | −0.002 (−0.08) | −0.001 (−0.05) | −0.756 *** (−2.51) | 0.013 (0.55) |
| GOI | 0.038 (0.93) | 0.046 (1.15) | 0.05 (1.17) | 0.049 (1.2) | 0.5 (0.85) | 0.06 (1.37) |
| POP | 0.279 (0.76) | 0.287 (0.77) | 0.88 ** (2.12) | 0.88 ** (2.23) | 1.28 (0.22) | 0.96 ** (2.36) |
| FTD | 0.077 (1.27) | 0.057 (0.89) | 0.31 *** (2.72) | 0.304 *** (2.01) | −0.796 (−0.49) | 0.357 *** (3.07) |
| KAC | 0.124 ** (2.33) | 0.15 *** (2.71) | 0.109 ** (1.94) | 0.108 ** (2.01) | 0.195 (0.24) | 0.119 ** (2.06) |
| $CO_2$ | 0.122 * (1.7) | 0.127 * (1.72) | 0.022 (0.27) | 0.022 (0.28) | 0.093 (0.07) | 0.03 (0.32) |
| $W^*$EPC | | | −0.033 (−0.29) | −0.033 (−0.31) | −0.019 (−0.01) | −0.126 (−0.89) |
| $W^*$EPE | | | −0.18 ** (−1.91) | −0.169 * (−1.87) | −0.24 (−0.2) | −0.101 (−1) |
| $W^*$PMS | | | 0.037 (0.96) | 0.037 (1.02) | −0.055 (−0.1) | 0.068 * (1.71) |
| $W^*$GOI | | | −0.088 (−1.31) | −0.087 (−1.36) | 0.134 (0.14) | −0.065 (−0.94) |
| $W^*$POP | | | −0.262 (−0.38) | −0.242 (−0.37) | −2.247 (−0.22) | 0.086 (0.12) |
| $W^*$FTD | | | 0.503 *** 3.64) | 0.507 ***. (3.85) | 0.072 (0.02) | 0.324 (1.4) |
| $W^*$KAC | | | 0.246 *** (2.9) | 0.246 *** (3.06) | 0.899 (0.68) | 0.187 ** (1.99) |
| $W^*CO_2$ | | | 0.095 (0.83) | 0.096 (0.88) | −0.67 (−0.23) | 0.143 (0.69) |
| $\log l$ | 563.55 | 564.69 | 493.98 | 579.73 | 52.2 | 584.63 |

*, **, *** mean that *p* values are less than 0.1, 0.05, and 0.01, respectively. All tables below are the same. The data were obtained by Stata 15.0.

All results demonstrate that in the spatial fixed-effect model, it is better to choose the spatial adjacent weight matrix to analyze the spatial spillover effect, and in the spatial-temporal fixed-effect model, it is better to choose the spatial geospatial distance matrix. Furthermore, to analyze the spatial spillover effect, it is better to choose the spatial-temporal fixed-effect model with the geospatial distance matrix than the spatial fixed-effect model with the adjacent matrix.

The explanatory variables in the space fixed-effect model and the spatial-temporal fixed-effect model revealed that the voluntary regulations were positively correlated with the SDC of Chinese ports and the EPE was positive and statistically significant, therefore strengthening that voluntary regulation can improve the SDC of Chinese ports. Neither

EPC nor PMS was statistically significant, meaning neither mandatory regulation nor public media regulation could help a port to improve its SDC.

**Table 9.** Regressional results by spatial econometric models with $W_{ij}^S$.

| Variable | SAR | SEM | SDM | | | |
| --- | --- | --- | --- | --- | --- | --- |
| | | | No Fixed | Spatial Fixed | Time Fixed | Spatial-Temporal Fixed |
| EPC | −0.051 (−0.74) | −0.067 (−0.94) | 0.028 (0.36) | 0.028 (0.37) | −0.352 (−0.31) | −0.046 (−0.59) |
| EPE | 0.467 *** (11.33) | 0.478 *** (11.5) | 0.487 *** (11.79) | 0.486 *** (12.4) | 0.708 (1.15) | 0.483 *** (11.49) |
| PMS | 0.022 (0.94) | 0.021 (0.91) | 0.003 (0.15) | 0.004 (0.18) | −0.876 *** (−2.73) | 0.014 (0.61) |
| GOI | 0.032 (0.8) | 0.039 (0.95) | 0.041 (1.26) | 0.04 (1.02) | 0.538 (0.91) | 0.075 * (1.84) |
| POP | 0.312 (0.85) | 0.268 (0.72) | 1.116 *** (2.56) | 1.119 *** (1.26) | −0.399 (−0.06) | 0.704 * (1.64) |
| FTD | 0.079 (1.3) | 0.058 (0.86) | 0.327 *** (2.84) | 0.324 *** (2.96) | −0.444 (−0.25) | 0.251 ** (2.06) |
| KAC | −0.122 ** (−2.28) | −0.136 ** (−2.47) | 0.075 (−1.33) | 0.074 (−1.38) | 0.115 (0.14) | 0.137 ** (−2.41) |
| $CO_2$ | 0.125 * (1.73) | 0.124 * (1.67) | 0.01 (0.09) | 0.01 (0.11) | 1.066 (0.68) | 0.19 * (1.75) |
| $W^*$EPC | | | 0.131 (0.68) | 0.135 (0.74) | −2.349 (−0.48) | −0.597 * (−1.78) |
| $W^*$EPE | | | −0.245 (−1.52) | −0.23 (−1.49) | −1.049 (−0.36) | −0.21 (−0.94) |
| $W^*$PMS | | | 0.077 (1) | 0.076 (1.04) | −1.144 (−0.87) | 0.168* (1.81) |
| $W^*$GOI | | | −0.253 ** (−2.14) | −0.254 ** (−2.26) | 0.182 (0.09) | −0.123 (−0.91) |
| $W^*$POP | | | 0.377 (0.34) | 0.394 (0.38) | 2.88 (0.15) | 0.539 (0.41) |
| $W^*$FTD | | | 0.638 *** (3.89) | 0.643 *** (4.13) | 3.076 (0.42) | 0.914* (1.84) |
| $W^*$KAC | | | 0.374 ** (2.23) | 0.377 ** (2.38) | 1.966 (0.61) | 0.098 (0.44) |
| $W^*CO_2$ | | | 0.142 (1.01) | 0.145 (1.09) | 6.741 (0.92) | 1.351 *** (2.66) |
| $\log l$ | 563.73 | 563.77 | 493.92 | 580.24 | 47.97 | 586.46 |
| $R^2$ | 0.446 | 0.441 | 0.528 | 0.528 | 0.0064 | 0.145 |
| Obs | 200 | 200 | 200 | 200 | 200 | 200 |

*, **, *** mean that *p* values are less than 0.1, 0.05, and 0.01, respectively. All tables below are the same. The data were obtained by Stata 15.0.

Among the control variables, market size, foreign trade dependence, and knowledge accumulation capacity were positively correlated with the SDC of Chinese ports in both two fixed-effect models based on three weight matrices. POP, FTD, and KAC were positive and statistically significant, suggesting that ports can improve their SDC by expanding their market size, increasing international trade, or recruiting more top talent. GOI only passed the significant test in the spatial-temporal fixed-effect model, and $CO_2$ was not statistically significant in any model.

**Table 10.** Regressional results by spatial econometric models with $W_{ij}^E$.

| Variable | SAR | SEM | SDM | | | |
|---|---|---|---|---|---|---|
| | | | No Fixed | Spatial Fixed | Time Fixed | Spatial-Temporal Fixed |
| EPC | −0.054<br>(−0.79) | −0.073<br>(−1.05) | −0.016<br>(−0.2) | −0.012<br>(−0.16) | −0.28<br>(−0.29) | 0.01<br>(0.07) |
| EPE | 0.468 ***<br>(11.51) | 0.47 ***<br>(11.66) | 0.5 ***<br>(11.46) | 0.5 ***<br>(12.06) | 0.586 **<br>(1.19) | 0.481 ***<br>(11.9) |
| PMS | 0.022<br>(0.95) | 0.021<br>(0.92) | 0.021<br>(0.85) | 0.022<br>(0.94) | −0.7 ***<br>(−2.54) | 0.026<br>(1.08) |
| GOI | 0.027<br>(0.66) | 0.026<br>(0.66) | 0.053<br>(1.26) | 0.052<br>(1.3) | −0.0005 ***<br>(−3.32) | 0.08 **<br>(1.95) |
| POP | 0.319<br>(0.87) | 0.392<br>(1.08) | 0.051<br>(1.24) | 0.49<br>(1.26) | 0.418<br>(0.83) | 1.09**<br>(2.41) |
| FTD | 0.072<br>(1.2) | 0.065<br>(1.01) | 0.167<br>(1.36) | 0.164<br>(1.4) | 0.387<br>(0.07) | 0.33***<br>(2.64) |
| KAC | −0.12 **<br>(−2.26) | −0.13 **<br>(−2.34) | 0.113 **<br>(1.89) | 0.115 **<br>(2.03) | 1.02<br>(0.67) | 0.053<br>(0.89) |
| $CO_2$ | 0.13 *<br>(1.82) | 0.13 *<br>(1.72) | 0.064<br>(0.65) | 0.066<br>(0.7) | 0.171<br>(0.14) | 0.05<br>(0.49) |
| $W^*$EPC | | | 0.232<br>(1.14) | 0.244<br>(1.26) | −0.106<br>(−0.02) | 0.535<br>(1.4) |
| $W^*$EPE | | | 0.225 *<br>(1.55) | 0.169 **<br>(1.19) | 1.02<br>(0.69) | 0.226 **<br>(1.42) |
| $W^*$PMS | | | 0.01.<br>(0.22) | 0.01<br>(19) | 0.036<br>(0.07) | 0.04<br>(0.91) |
| $W^*$GOI | | | 0.071<br>(0.81) | 0.07<br>(0.85) | 0.154<br>(0.14) | 0.117<br>(1.3) |
| $W^*$POP | | | −1.808*<br>(−1.86) | −1.836 **<br>(−1.99) | 0.953<br>(0.06) | 0.672<br>(0.51) |
| $W^*$FTD | | | 0.247<br>(1.33) | 0.253<br>(1.43) | −1.224<br>(−0.32) | −0.394<br>(−1.25) |
| $W^*$KAC | | | −0.002<br>(−0.02) | −0.003<br>(−0.03) | −0.121<br>(−0.09) | −0.01<br>(−0.07) |
| $W^*CO_2$ | | | 0.05<br>(0.31) | 0.053<br>(0.37) | −1.995<br>(−0.52) | −0.156<br>(−0.49) |
| log $l$ | 564.65 | 565.38 | 488.1 | 572.13 | 70.7 | 582.2 |
| $R^2$ | 0.441 | 0.441 | 0.474 | 0.477 | 0.0012 | 0.0462 |
| Obs | 200 | 200 | 200 | 200 | 200 | 200 |

*, **, *** mean that $p$ values are less than 0.1, 0.05, and 0.01, respectively. All tables below are the same. The data were obtained by Stata 15.0.

### 4.3.3. Spatial Spillover Effect

When a spatial lag exists in the SDM, its regressional result will not directly embody the impact of explanatory variables on the SDC. Based on a method where LeSage and Pace [62] solve this problem by decomposing the total effects into direct and indirect effects, the effects of variables were divided into direct effects, indirect effects, and total effects, and several partial differential equations were designed to evaluate the spatial spillover effect. The spatial spillover effect of Chinese Ports was investigated by the spatial fixed-effect model with $W_{ij}^A$ and the spatial-temporal fixed-effect model with $W_{ij}^S$, respectively. The results are shown in Tables 11 and 12.

**Table 11.** The spatial spillover effect by the spatial fixed-effect model with $W_{ij}^A$.

| SDM | Variable | Effect of Type | | |
|---|---|---|---|---|
| | | **Direct Effect** | **Indirect Effect** | **Total Effect** |
| Spatial Fixed | IEPC | 0.034 (0.48) | −0.036 (−0.29) | −0.001 (−0.01) |
| | EPE | 0.51 *** (12.05) | −0.116 (−1.34) | 0.394 *** (3.69) |
| | PMS | 0.0003 (0.01) | 0.042 (1) | 0.043 (0.87) |
| | GOI | 0.046 (1.08) | −0.082 (−1.11) | −0.035 (−0.36) |
| | POP | 0.853** (2.22) | −0.15 (−0.21) | 0.703 (0.86) |
| | FTD | 0.27 *** (2.75) | 0.506 *** (4.16) | 0.237 ** (2.29) |
| | KAC | 0.263 *** (2.84) | 0.096 * (1.85) | 0.167 * (1.5) |
| | $CO_2$ | 0.03 (0.4) | 0.108 (0.88) | 0.138 (1.07) |

*, **, *** mean that *p* values are less than 0.1, 0.05, and 0.01, respectively. All tables below are the same. The data were obtained by Stata 15.0.

**Table 12.** The spatial spillover effect by the spatial-temporal fixed-effect model with $W_{ij}^S$.

| SDM | Variable | Effect of Type | | |
|---|---|---|---|---|
| | | **Direct Effect** | **Indirect Effect** | **Total Effect** |
| Spatial-temporal fixed | EPC | −0.037 (−0.48) | −0.513 * (−1.66) | −0.55 * (−1.74) |
| | EPE | 0.488 *** (11.44) | −0.269 * (−1.45) | 0.219 *** (1.06) |
| | PMS | 0.011 (0.49) | 0.144 * (1.75) | 0.155 * (1.78) |
| | GOI | 0.078 ** (1.88) | −0.108 (−0.9) | −0.03 (−0.23) |
| | POP | 0.675 * (1.59) | 0.342 (0.3) | 1.016 (0.86) |
| | FTD | 0.255 ** (2.25) | 0.835 (1.84) | 0.58 ** (1.15) |
| | KAC | 0.141*** (2.6) | 0.12 (0.58) | 0.212 * (1.3) |
| | $CO_2$ | 0.174 ** (1.69) | 1.175 ** (2.45) | 1.349 ** (2.5) |

*, **, *** mean that *p* values are less than 0.1, 0.05, and 0.01, respectively. All tables below are the same. The data were obtained by Stata 15.0.

The results revealed that all spatial econometric models were robust to study the SDC spillover effects of the Chinese ports. Among those models, the spatial-temporal fixed-effect model with the geospatial distance matrix was better in studying it than the spatial fixed-effect model with the adjacent matrix.

4.3.4. Discussion of Spatial Spillover Effects

The direct effect and the total effect of EPE on the SDC passed the 1% significance test positively, which means that voluntary regulation helps to improve the port's SDC and its competitiveness. The indirect effect of EPE on the SDC passed the 10% significance test negatively, which means that other ports will compete for enhancement of their SDC after a port adopts more voluntary regulations.

The direct effect of GOI on the SDC passed the 5% significance test positively, and the indirect effect and the total effect of GOI on the SDC were negative, which failed the significance test. It can be concluded that expanding a port's operation scale will improve its SDC, thereby, it will raise its revenue and gain more competitiveness. However, it may hurt the other ports' operations, which will weaken the SDC of other ports. When a port expands its operation scale as a monopoly, it will harm the competitiveness of the entire port industry.

The direct effect of POP on the SDC passed the 10% significance test positively, and the indirect effect and the total effect of POP on the SDC were positive but failed the significance test. It can be concluded that the growth of population in a port will add

workforce and attract top talent, and then improve the port's SDC. Other ports may rely on the population growth to improve their SDC, but this effect is not significant.

The direct effect and the total effect of FTD on the SDC passed the 5% significance test positively, and the indirect effect of FTD on the SDC was positive but failed the significance test. This reveals that the economic development of a port will improve its SDC and may help other ports to improve their SDC and the nationwide SDC, but the latter effect is not significant.

The direct effect of KAC on the SDC passed the 5% significance test positively, and the indirect effect and the total effect of KAC on the SDC were positive but failed the significance test. It can be concluded that a port recruiting more top talent will improve its SDC, and other ports will do the same thing to improve their SDC, but the latter effect is not significant.

The direct effect, the indirect effect, and the total effect of $CO_2$ on the SDC passed the 5% significance test positively. It is generally accepted that improving the SDC relies on expanding the port's cargo volume, which will add carbon emissions and cause more environmental problems. There are three ways to control carbon emissions: (1) a port should strengthen its voluntary regulations; (2) the port authority should enact more effective ERs; and (3) the public media as a supervision tool could focus on the environmental problems.

## 5. Conclusions and Suggestions

In this paper, a two-stage WP-SBM-DEA model was constructed to address the dynamic operational features of Chinese ports and to investigate the spatial characteristics of their SDC, whose data were from 2009 to 2018. The spatial spillover effects of various ERs on the port's SDC are discussed, which revealed the synergistic effects of various ERs on the port's SDC and suggests that the port authority and port enterprises rethink the importance of ERs.

After investigating the SDC of Chinese ports, we showed that the SDC of Chinese ports varies by location, and they are heterogeneous and clustered spatially. After comparing several DEA models, an academic finding revealed that the spatial-temporal fixed-effect model with the geospatial distance matrix was more suitable to investigate the spatial effects of the port's SDC. The results of the spatial spillover effect study illustrates that a powerful tool for the port companies to balance the economic development and ecological civilization is to adopt more voluntary regulations than any other regulations. The possible measures to boost a port's SDC are to expand its operation scale and market size, to increase its international market, and to recruit more top talent.

There are also several suggestions for the port authorities and companies. Since the spatial spillover effect of the port's SDC is mostly affected by the geospatial distance, the port authority should strengthen the mutual water transportation between ports. Since the advanced ports, mainly located in the Bohai Rim region, the Yangtze River Delta region, and the Pearl River Delta region, have higher cargo volume and emit more carbon dioxide than others, the port authority should enact different ERs to improve the port's SDC in terms of its environmental pressure. After comparing three kinds of regulations, an excessive voluntary regulation or public media regulation would hurt the port's SDC. Therefore, the implementation of ERs and regulations should conform to the principle of appropriateness and local conditions.

Future research should pay attention to the synergy of green technology on the port's SDC. For example, introducing clean energy technology for the port equipment, as a frontier in the port industry, would propel the improvement in the port's SDC. In addition, more mathematical models can be developed to quantitatively evaluate the synergy effects of ERs on the port's SDC.

**Author Contributions:** Conceptualization, X.H. and W.H.; Methodology, X.H., W.L., R.H., and W.H.; Software, W.L. and R.H.; Validation, X.H., W.L., R.H., and W.H.; Formal analysis, W.L., R.H., and W.H.; Investigation, W.L., R.H., and W.H.; Resources, X.H., W.L., and R.H.; Data curation, R.H. and W.H.; Writing—original draft preparation, X.H., W.L., and R.H.; Writing—review and editing, X.H.,

W.L., R.H., and W.H.; Visualization, W.L. and R.H.; Supervision, X.H.; Project administration, X.H. and W.H.; Funding acquisition, W.H and X.H. All authors have read and agreed to the published version of the manuscript.

**Funding:** This research was funded by the National Natural Science Foundation of China (NSFC) (grant numbers 71971158, 71371145, and 71473162).

**Institutional Review Board Statement:** Not applicable.

**Informed Consent Statement:** Not applicable.

**Data Availability Statement:** Data is contained within the article.

**Acknowledgments:** The authors would like to thank all supporters and commenters in this research.

**Conflicts of Interest:** The authors declare no conflict of interest.

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
