# Peer review of "Environmental Regulations on the Spatial Spillover of the Sustainable Development Capability of Chinese Clustered Ports"

_jmse, doi:10.3390/jmse9030301_

Round 1
Reviewer 1 Report
Dear authors,
please, see the attachment.
Best regards

Author Response
Point 1: Introduction. This section contains several claims that are not supported by any reference, e.g. "The development of ports is becoming a comprehensive indicator to measure the country's competitive level". In addition, more importantly, you should better single out the scientific gap(s) that this work aims at filling and why it may be worthwhile to do so from both a scientific and a practical perspective. Currently, this information is absent. Such a lack is rather critical, since it justifies the manuscript. How did you set your objective? And why is a scientific paper need for tackling it? Also, the SDC concept is pretty vague. When you refer to the Porter's Hypothesis, which aspect are you considering? Profitability? Competitiveness? Innovation capability? Try to be as clear as possible, without any source of vagueness.
Response 1: Several references are added to support the claims in “1. Introduction”. The research gap and relevant problems are expressed in the Response 2.
Point 2: Literature review. The purpose of a literature review is to provide a literature background to better delimit the research gap. Instead, the current version of this Section seems a list of papers discussed without an overarching rationale. Thus, you should spend more effort towards this direction. I suggest:
o Introducing the port sustainability concept from several perspectives (environmental, social, economic) to provide introductory material. With this aim, the following reference may be useful:
Zerbino et al. (2019), "Towards analytics-enabled efficiency improvements in maritime transportation: A case study in a mediterranean port", Sustainability, 11(16), 4473.
o Moving Section 2.2 to the methodological section.
o Expanding 2.3 and 2.4 to better motivate why you should fulfill your research objective.
Response 2: “2. Literature review” is reorganized and rewritten, and its sections are extended to “2.1. Port’s SDC; 2.2. Effects of ERs on port’s SDC; 2.3. Cluster of ports; 2.4. Spatial spillover of the port’s SDC; 2.5. Methods for studying the port’s SDC; 2.6. Research gap”.
Meanwhile, the relevant bodies are added, and references are extended to 63 papers, which are cited properly.
Point 3: Methodology. The methodological section is rather sound and I appreciate the level of detail. Yet, you did not provide any justification for your methodological choices. This is a critical weak spot. You must motivate them by supporting your statement by appropriate references.
Response 2: (1) An introduction paragraph is inserted at the beginning of “3.1.1. The SDC evaluation model”, and Figure 1 is explained in detail. (2) The data in Table 3 are all adjusted by the date 2009 to 2018. (3) Results of LISA analysis are added as Figures 2 and 3, as well as relevant bodies.
Point 4: Discussion and Conclusions. You undoubtedly provided a few interesting hints stemming from the results, which is positive. Nonetheless, research is about contribution, which should be the most captivating part of a scientific work. The current contributions of the work are quite poor. Instead, I believe you may further reflect on your results to elicit compelling arguments. You should put extra effort in:
o Describing the scientific contributions of your work any specifying which research stream(s) you are tackling;
o Describing the managerial and policy contributions;
o Trying to generalize your findings;
o Detailing limits and further development of your manuscript;
o Expanding the interesting hint pertaining to the future introduction of clean energy technology and to why this may be useful.
Response 4: “5. Conclusions and suggestions” is rewritten.
Point 5: My overall impression is that this work may provide an appreciable contribution, but some major weaknesses are hampering its potential. In my humble opinion, you should not underestimate all the above-mentioned aspects.
Response 5: All references are cited properly and their academic contributions are respected in this paper.

Reviewer 2 Report
I think the core of the paper is figure 1, but I have a hard time understanding what this means. We ask for a re-edit as to why we should do this.
Thesis is not a list of experimental data. The key thesis should be explained earlier and the calculation data should be processed in the appendix.
Author Response
Point 1: I think the core of the paper is figure 1, but I have a hard time understanding what this means. We ask for a re-edit as to why we should do this.
Response 1: An introduction paragraph is inserted at the beginning of “3.1.1. The SDC evaluation model”, and Figure 1 is explained in detail and its context is rewritten.
Point 2: Thesis is not a list of experimental data. The key thesis should be explained earlier and the calculation data should be processed in the appendix.
Response 2: Previous researches are carefully reviewed and cited in the Literature review. The data in Table 3 are all adjusted by the date 2009 to 2018, and results of LISA analysis are added as Figures 2 and 3, as well as relevant bodies. We will move the calculation data to the appendix as you wish, we need more days to finish it, thank you.

Round 2
Reviewer 1 Report
Dear authors,
I praise you for the changes applied to the manuscript. In my humble opinion, you successfully coped with all my requests. Thus, I recommend an acceptance.
Best regards
Author Response
- Have Replaced “desirable” by “desired”, and “undesirable” by “undesired” throughout this paper.
- Our research data are retrieved from so many yearbooks, including China Statistical Yearbooks, China Environmental Yearbooks, and China’s Port Statistical Yearbooks from 2009 to 2018, that they are not listed in references.
- A reference [34] is inserted, and the later reference number adds one.
- Some words are modified according to your suggestions and marked red.
Wenfa Hu

Reviewer 2 Report
The author corrected what the reviewer pointed out.
Author Response

(The authors gave the same response as above.)
